# MuseTalk: Real-Time High Quality Lip Synchronization with Latent Space Inpainting

## Abstract

Achieving high-resolution, identity consistency, and accurate lip-speech synchronization in face visual dubbing presents significant challenges, particularly for real-time applications like live video streaming. We propose MuseTalk, which generates lip-sync targets in a latent space encoded by a Variational Autoencoder, enabling high-fidelity talking face video generation with efficient inference. Specifically, we project the occluded lower half of the face image and itself as an reference into a low-dimensional latent space and use a multi-scale U-Net to fuse audio and visual features at various levels. We further propose a novel sampling strategy during training, which selects reference images with head poses closely matching the target, allowing the model to focus on precise lip movement by filtering out redundant information. Additionally, we analyze the mechanism of lip-sync loss and reveal its relationship with input information volume. Extensive experiments show that MuseTalk consistently outperforms recent state-of-the-art methods in visual fidelity and achieves comparable lip-sync accuracy. As MuseTalk supports the online generation of face at 256x256 at more than 30 FPS with negligible starting latency, it paves the way for real-time applications. The codes and models will be made publicly available upon acceptance.

## 1 Introduction

Generating realistic talking face animations from audio has gained significant attention in recent years, with applications spanning visual dubbing, the film industry, digital assistants, and beyond (Kim et al., 2018; Pataranutaporn et al., 2021; Song et al., 2019; Gu et al., 2019). The primary goal of talking face generation is to synchronize two distinct modalities—audio and visual—so that the lip's movements align with the input speech content, producing lip-synced, high-fidelity videos.

Existing talking face generation approaches can be broadly classified into three main types based on their training paradigms and data requirements: *person-specific talking face* (Song et al., 2020; Lahiri et al., 2021; Guo et al., 2021), *one-shot talking head* (Stypu kowski et al., 2024; Chen et al., 2024; Xu et al., 2024a;b), and *few-shot face visual dubbing* (Cheng et al., 2022; Prajwal et al., 2020a; Zhong et al., 2023; Park et al., 2022; Wang et al., 2023). For *person-specific talking face* methods, the training process typically incorporates the subject's specific identity. While these methods can generate highly photorealistic talking face videos, the need for re-training or fine-tuning for each new speaker limits their practicality in real-world applications. Meanwhile, with the rapid advancements in GANs and diffusion models for video synthesis, *one-shot talking head* techniques have emerged, which can drive a single reference facial image to produce synchronized lip movements, realistic facial expressions, and natural portrait animations. Although these methods can generate vivid talking head videos, they require vast amounts of training data, significant computational resources, and time-consuming inference processes (Guo et al., 2024), making them unsuitable for real-time interactions, such as digital human live streaming. As a result, *few-shot face visual dubbing* techniques, which focus on reconstructing the mouth region of the source face based on driving audio, have gained significant attention in applications such as video translation, film dubbing, and real-time interactions like virtual digital humans, where speed and efficiency are crucial. But in fact, because humans are particularly sensitive to even slight misalignments between speech and facial movements, the key criteria for high-fidelity face visual dubbing are *high resolution*, *identity consistency*, and *lip-speech synchronization*. However, due to the weak correlation between audio and visual inputs, the primary challenge remains how to align these two heterogeneous modalities

sufficiently. Then, from an information-theoretic perspective, the central issue is to *effectively minimize the uncertainty (entropy) of the sources while mitigating the impact of noise and irrelevant information* (Lahat et al., 2015).

To achieve this purpose, some approaches typically leverage an encoder-decoder structure to align audio and visual representations. They use convolutional networks with several up-sampling layers to synthesize mouth region pixels directly from latent embeddings (Prajwal et al., 2020b; Xie et al., 2021; Cheng et al., 2022; Wang et al., 2023). Nevertheless, within such a straightforward fusion mechanism, the information screening process often fails to adequately preserve the facial texture details from the reference image. This can lead to a loss of texture quality and result in blurry, identity-inconsistent visuals, even with high-resolution training data (Zhang et al., 2023). Beyond visual quality, lip-speech synchronization is a more crucial metric for visual dubbing, as it measures the temporal correspondence between the input audio and the synthesized video streams. A straightforward yet effective approach to this issue is to incorporate an auxiliary pre-trained network, such as a lip-sync discriminator (Prajwal et al., 2020b; Chung & Zisserman, 2017) or a lip-sync loss (Park et al., 2022) at the end of the generator to assess audio-visual coherence. This can be viewed as a form of Mutual Information Estimator (Zhu et al., 2020), but there is limited research analyzing its mutual information regulation mechanism, particularly regarding how it selects the most relevant features from redundant source information. Therefore, this paper addresses the three key challenges of *few-shot face visual dubbing* by focusing on controlling the complexity and information volume of the input source, along with implementing an efficient information fusion mechanism. We also attempt to answer two fundamental questions: *(1) How to achieve high-resolution and Identity-Consistency visual content?* and *(2) How to obtain lip-synced mouth movement?*

In this paper, we introduce MuseTalk, a real-time framework for face visual dubbing that employs mouth inpainting technology. MuseTalk takes as input a occluded lower half of a face image, a reference face from the same individual, and an audio track, and outputs a face image with lips that are seamlessly synchronized with the audio. Specifically,

1. To generate high resolution face images ($256 \times 256$), while ensuring real-time inference capabilities, we introduce a method to produce lip-sync targets within a latent space. This space is encoded by a pre-trained Variational Autoencoder (VAE) Kingma & Welling (2013), which is instrumental in maintaining the quality and speed of our framework.

2. Within the latent space, we employ a U-Net structure for the generative model. This structure integrates visual and audio embeddings across different scales through cross-attention. This approach allows MuseTalk to manage the flow of information efficiently, preserving essential visual features and enhancing the synchronization.

3. We observe that, during testing, the closer the pose of the occluded face is to that of the reference face during testing, the better the generation results, due to reduced variation. Typically, both the occluded and reference faces are derived from the same frame, akin to the approaches in Cheng et al. (2022); Prajwal et al. (2020b). However, during training, they adopt a random sampling strategy that introduces significant pose variations between the occluded and reference faces. To bridge this gap, we propose Selective Information Sampling (SIS), which selects reference images with head poses closely aligned to the target while ensuring distinct lip movements. This strategy sharpens the model's focus on the intricate textures of the mouth region.

4. We also delve into the mechanism of the widely utilized lipsync loss by using our proposed *Adaptive Audio Modulation*(AAM) strategy to understand its role in enhancing lip-sync accuracy. By modulating the amount of input information, the lip-sync loss facilitates the model's ability to extract more pertinent information from the two modalities, thereby optimizing the mutual information.

We conduct qualitative and quantitative experiments to evaluate our MuseTalk, and experimental results show that our method significantly outperforms existing techniques in terms of both visual quality and lip-sync accuracy, paving the way for more advanced applications in audio-driven visually dubbing and beyond.

The remainder of this paper is organized as follows: Sec.2 presents related work in the field. Section 3 describes the main framework of MuseTalk. Section 4 provides experimental results, and Section 5 concludes the paper while discussing limitations and future directions.

## 2 RELATED WORK

Existing audio-driven talking face generation methods can be broadly categorized into three types: *person-specific*, *one-shot talking head* and *few-shot face visual dubbing* methods. The key difference among them is how much information do they need from reference visual content to generation of photo-realistic talking face animations.

**Person-specific talking face** methods, which incorporate target characteristics during training, can generate high-fidelity, identity-preserving results. However, these approaches typically rely on 3D models (Song et al., 2020; Thies et al., 2020; Guo et al., 2021) or Neural Radiance Fields (NeRF) (Park et al., 2022) as intermediate representations and require several minutes of footage to learn the audio-lip mapping, making them time-consuming and limiting their real-world applicability.

**One-shot talking head** methods have recently gained significant attention due to their ability to generate lifelike facial expressions and manage head motion dynamics using a single reference image. A common approach is to use intermediate representations like facial landmarks or 3D meshes. For instance, (Chen et al., 2019; Zhou et al., 2020) propose a two-stage pipeline where an audio-to-landmark module is followed by a landmark-to-video generation, effectively disentangling speaker identity from speech content. Other works, such as (Chen et al., 2020; Zhang et al., 2021a), utilize 3D coefficients to drive facial motion based on audio-predicted expression, pose, and geometry. However, these methods often struggle with fine-grained details like teeth and mouth textures, limiting the overall fidelity of the generated videos due to the coarse control granularity of 3D models. With the recent advancements in diffusion models for video generation, audio-driven talking head synthesis has seen progress with single-image setups. Works such as (Tian et al., 2024; Xu et al., 2024a; Chen et al., 2024; Wang et al., 2024; Xu et al., 2024b) demonstrate how diffusion models can generate realistic portrait videos driven by audio inputs. While diffusion methods excel in enhancing generation diversity, they can introduce uncertainty and loss of identity-related details, leading to artifacts such as overly beautified faces or loss of specific textures. Furthermore, these methods require large amounts of training data, substantial computational resources, and a time-intensive multi-step inference process, which limits their practicality for real-time or resource-constrained applications.

**Few-shot face visual dubbing** focuses on replacing the mouth region of a source face based on driving audio. The most common approach utilizes an encoder-decoder architecture. For instance, works like (Prajwal et al., 2020a; Park et al., 2022; Xie et al., 2021; Cheng et al., 2022) employ separate image and audio encoders to extract features, which are then fused using a single decoder to generate the mouth region pixels directly on the source face. While these methods achieve lip movements that correspond well to speech content, the simple fusion mechanism often fails to preserve the facial texture details from the reference image. This results in a loss of texture fidelity and produces blurry, identity-inconsistent visual outputs, even when trained on high-resolution data (Zhang et al., 2023). To address these limitations, (Zhang et al., 2023) propose a Deformation Inpainting Network, where a deformation operation shifts pixels into their correct positions, preserving high-frequency texture details. However, this method can still introduce local blur due to redundant information from the reference images and restrict natural lip movement. Additionally, the inpainting module is prone to overfitting, leading to noticeable color differences when applied to unseen images.

## 3 METHOD

In this section, we present the details of the MuseTalk framework, as shown in Fig.1. First, we provide an overview of the network architecture, covering the inputs, outputs, and key components of MuseTalk. Next, we delve into the auxiliary training strategies from the perspective of information modulation, specifically the *Selective Information Sampling* (SIS) and *Adaptive Audio Modulation* (AAM) methods. Finally, we outline the implementation details for both the training and testing phases of MuseTalk.

## 3.1 FRAMEWORK

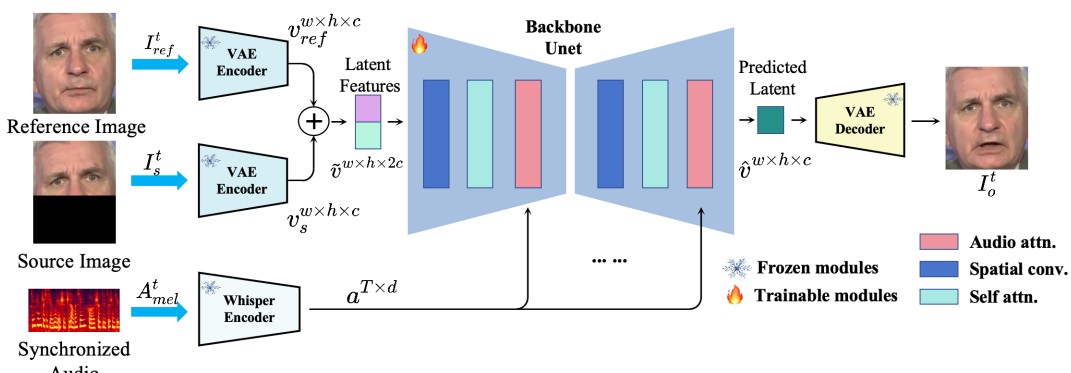

Figure 1: Illustration of our MuseTalk framework. We first encodes a reference facial image and a occluded lower half target image into perceptually equivalent latent space. Subsequently, we employ a multi-scale learning network structure to effectively fuse audio and visual features at various scales, thereby facilitating a more comprehensive integration of the two modalities. Consequently, the decoded results from the latent space yield more realistic and lip-synced talking face visual content.

MuseTalk is an innovative framework designed for multi-scale modality alignment, focusing on the synchronization of audio and visual elements. Our network structure is inspired by the Latent Diffusion Model (LDM)(Rombach et al., 2022), which employs a pretrained autoencoder (VQ-VAE(Van Den Oord et al., 2017)) to map images from pixel space to latent space, where the diffusion process is performed. The training objective is formulated as $L = \mathbb{E}_{z_t, c, \varepsilon \sim \mathcal{N}(0,1), t}[\|\epsilon - \epsilon_\theta(z_t, t, c)\|_2^2]$, where $\epsilon_\theta$ represents the backbone denoising network, including several condition-related cross attention modules. As demonstrated in (Rombach et al., 2022), operating in latent space promotes local realism and avoids the blurriness that often arises from pixel-space losses, such as $L_2$ or $L_1$ objectives. Building on this, we adopt a similar mechanism from Stable Diffusion, a text-to-image diffusion model based on LDM. While the diffusion mechanism enhances diversity by bridging the conditional and target domains through noise infusion, it also introduces greater uncertainty in the generated outputs. For talking face generation, where identity consistency and lip synchronization are critical, it's essential to establish a more direct correlation between the fused features and the final results. Therefore, we make the following adjustments: (1) **we bypass the complex and time-consuming diffusion process, exporting the final result directly**; (2) instead of a single input image, we use an occluded lower half of the target image along with a reference facial image, while the driving condition is a sequence of audio embeddings

As shown in Fig. 1, an occluded lower half of the target image $I_s^t$ and a reference identity image $I_{ref}^t$ at time $t$ are each passed through a pre-trained VAE encoder. The resulting outputs $v_{ref}^{w \times h \times c}$ and $v_m^{w \times h \times c}$ are then concatenated along the channel dimension to create a comprehensive image feature representation $v^{w \times h \times 2c}$, where $w$ and $h$ denote the width and height of the feature. For the audio data, we leverage a pre-trained Whisper (Radford et al., 2023) encoder to extract features from a sequence audio segment. The length of the audio segment is set to $T$, centered at time $t$. This segment is first re-sampled to 16,000 Hz and then transformed into an 80-channel log magnitude Mel spectrogram, which serves as the input $A_d^t \in R^{T \times 80}$. The output audio feature shape is $a^{T \times d}$, where $d$ denotes the dimension of audio feature. The selection of value $T$ is crucial for capturing the temporal dynamics of the spoken content, which will be evaluate in Sec.3.3 especially in modulating the amount of input information.

Then, it is well understood that the crux of generating a realistic talking face lies in the alignment of the two heterogeneous modalities, audio and visual. Traditional approaches that merely use multiple up-sampling convolutional layers to directly generate mouth shape pixels fall short for creating high-quality, lip-synced videos. To address this, we draw inspiration from the success of the U-Net structure (Ronneberger et al., 2015), widely recognized in recent multi-modal generation tasks (Rombach et al., 2021). The U-Net's multi-scale learning network architecture adeptly fuses

audio and visual features across various scales with the capable of modeling conditional distributions of the the form $p\left(\hat{v} \mid a\right)$, enabling a more nuanced and holistic integration of the two modalities.

Ultimately, the fused feature $\hat{v}_t^{w \times h \times c}$ is fed into a pre-trained VAE decoder to generate the final results. This refined process ensures that the generated video not only aligns perfectly with the audio track but also maintains the visual coherence and quality expected in state-of-the-art audio-vision synchronization systems.

**Audio Encoder.** Accurate lip-synced talking face generation relies heavily on robust audio embeddings. In our approach, we utilize Whisper (Radford et al., 2023), a versatile, general-purpose speech recognition model, as our audio encoder due to its proven effectiveness in multilingual and multitask scenarios. We specifically use the encoder part, which processes the input via two convolution layers with GELU activation, adds sinusoidal position embeddings, and applies Transformer blocks with pre-activation residuals, followed by a final layer normalization.

**Loss Function.** As shown in Fig. 1, a synthesized talking face image $I_o^t$ and given a ground truth image $I_{gt}^t$, three loss functions are applied to improve the video generation quality, including reconstruction loss, perception loss (Johnson et al., 2016), GAN loss (Mao et al., 2017) and lip-sync loss (Prajwal et al., 2020b). We use L1 loss for reconstruction to minimize pixel-wise differences, ensuring color and structure consistency between generated and target images. However, it often misses fine textures. Perception loss, on the other hand, emphasizes perceptual similarity by comparing high-level features, enhancing details and visual realism. To balance both, we combine L1 and Perception loss for global consistency and fine detail preservation, as shown in Equation (1) and (2). Additionally, GAN loss (see Equation (5)) encourages the model to learn subtle details by challenging a discriminator. Together, L1, Perception, and GAN losses promote fidelity, realism, and perceptual quality in the generated images.

$$\mathcal{L}_{rec} = \left\| I_o^t - I_{gt}^t \right\|_1 \quad (1) \qquad \mathcal{L}_p = \left\| \mathcal{V}(I_o^t) - \mathcal{V}(I_{gt}^t) \right\|_2 \quad (2)$$

where $\mathcal{V}$ denotes the feature extractor of VGG19 (Simonyan & Zisserman, 2015).

$$\mathcal{L}_G = \mathbb{E}\left[ log(1 - D(I_o^t)) \right] \quad (3) \qquad \mathcal{L}_D = \mathbb{E}\left[ log(1 - D(I_{gt}^t)) \right] + \mathbb{E}\left[ log(D(I_o^t)) \right] \quad (4)$$

$$\mathcal{L}_{GAN} = \mathcal{L}_G + \mathcal{L}_D \quad (5)$$

In Equation (3) and (4), $\mathcal{L}_D$ optimizes the discriminator $D$ to distinguish synthesized face image $I_o^t$ from ground truth image $I_{gt}^t$, while $\mathcal{L}_G$ improves the result quality to fool the discriminator. Besides, as similar in (Prajwal et al., 2020a; Cheng et al., 2022; Zhang et al., 2023), we add a lip-sync loss to improve the synchronization of lip movements in dubbed videos. we also analyze the mechanism of lip-sync loss and reveal its relationship with input information volume in Sec.3.3. The lip-sync loss is defined in Equation (6), where our re-trained SyncNet (Prajwal et al., 2020a) takes $N$ pairs of audio and image frames as input. The output features are then used to calculate the cosine similarity with $P_{sync}$.

$$\mathcal{L}_{sync} = \frac{1}{N} \sum_i^N -log[P_{sync}(sycnet(A_{mel}^i, I_o^i))] \quad (6)$$

Finally, we weighted sum above losses as final loss $\mathcal{L}$,which is written as Equation (7), where we set $\lambda = 0.01$, $\mu = 0.01$ and $\varphi = 0.03$ in our experiment.

$$\mathcal{L} = \mathcal{L}_{rec} + \lambda \mathcal{L}_p + \mu \mathcal{L}_{GAN} + \varphi \mathcal{L}_{\text{sync}} \quad (7)$$

## 3.2 SELECTIVE INFORMATION SAMPLING(SIS)

To achieve high-resolution and identity-consistent visual results, it is crucial to retain relevant texture details while filtering out redundant information. To address this, we propose a Selective Information Sampling (SIS) strategy that selects reference images with head poses closely aligned to the target, while ensuring distinct lip movements, as shown in Fig. 2 (a).

First, we calculate the head pose similarity for each frame in the video using the euclidean distance between chin landmarks, identifying the *top-k* similar frames as the Pose-Aligned Image Set $\mathcal{E}_{pose}$. Next, we calculate the euclidean difference based on the inner-lip landmarks and identify the *top-k* frames with the most distinct lip movements, forming the Lip motion Dissimilarity Image

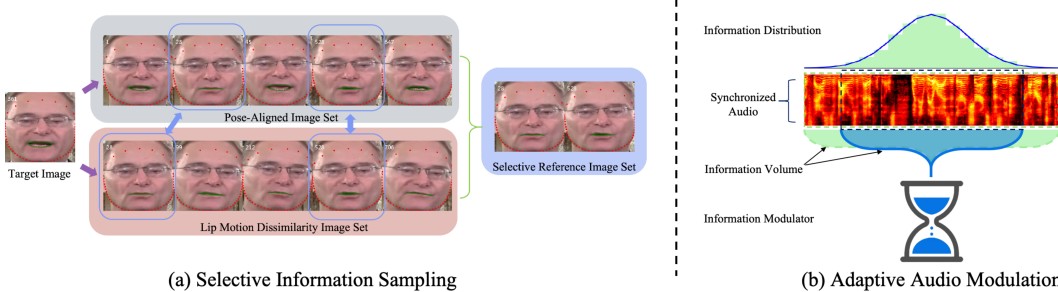

(a) Selective Information Sampling      (b) Adaptive Audio Modulation

Figure 2: The illustration of our proposed information modulation mechnism, including (a) Selective Information Sampling (SIS) and (b) Adaptive Audio Modulation (AAM).

Set $\mathcal{E}_{mouth}$. Finally, we select the intersection of the two sets $\mathcal{E}_{pose} \bigcap \mathcal{E}_{mouth}$ as the final Selective Reference Image Set $\mathcal{E}$ for each training sample. Unlike previous methods that randomly select reference images from the video sequence, our approach ensures that irrelevant and redundant information is removed, allowing the model to better focus on the detailed texture of the mouth region.

### 3.3 ADAPTIVE AUDIO MODULATION(AAM)

The driving audio is the key factor influencing the patterns of lip movements, so the information in both modalities (audio and visual) must be complete and highly coherent, capturing cues like pronunciation and tonality. The conventional approach is to manually synchronize audio and video before training. For example, (Wang et al., 2023) uses a 0.2-second audio segment centered on the pose reference as input. However, achieving precise synchronization is challenging due to different sampling rates between the two modalities. While our multi-scale fusion architecture addresses this issue to some extent, it does not entirely resolve it.

Building on previous research, we found that incorporating lip-sync loss (Prajwal et al., 2020a; Cheng et al., 2022; Zhang et al., 2023) significantly enhances lip synchronization. However, we sought to understand its underlying mechanism. Through experiments, we discover that lip-sync loss acts as an information modulator, regulating the input data and enabling the model to extract more relevant information from both modalities, thus optimizing mutual information. As illustrated in Fig. 2 (b), we assume the information contained in the synchronized audio input corresponding to the target lip movement follows a normal distribution. Lip synchronization-related information is concentrated in the middle range, while other information may pertain to finer details, such as pronunciation habits and temporal relations. To evaluate the quality of the generated results, we vary the length of the input audio segment ($T = 1, 3, 5, 7$) to control the information volume. This not only enhances lip-speech synchronization but also improves the image quality (see *Frechet Inception Distance (FID)* in Table 4).

## 4 EXPERIMENT

In this section, we describe the datasets and implementation details of our experiments. We first present a quantitative evaluation by comparing our method's performance with state-of-the-art approaches using relevant metrics. Next, we show qualitative results to highlight the visual fidelity of our synthetic outputs. Finally, we conduct ablation studies to evaluate the contributions of different components of our framework to its overall performance.

### 4.1 EXPERIMENTAL SETUP

**Implementation Details.** The training processes for MuseTalk are conducted on 2 NVIDIA H20 GPUs. The Unet model is initially trained with L1 loss and perceptual loss for 200,000 steps, which takes approximately 60 hours. Subsequently, the Unet is trained with lip-sync loss and GAN loss for an additional 100,000 steps, requiring around 30 hours. We use the whisper-tiny model [1] and

---

[1] https://github.com/openai/whisper

sd-vae-ft-mse [2] as audio and image feature extractors, respectively. As for image prepcocessing, we detect faces on each image as Region of interest (ROI) and then crop and resize the ROI to $256 \times 256$. The value of the $k$ in SIS is set to $40\%$ frames of the video length.

**Dataset Preparation.** We utilize two widely recognized high-resolution talking face datasets in our experiments: HDTF (Zhang et al., 2021b) and MEAD (Wang et al., 2020). The HDTF dataset consists of approximately 410 in-the-wild videos at 720P or 1080P resolution. We randomly select 20 videos for testing and use the remaining videos for training. All videos are trimmed to thirty-second clips for both the training and testing phases. We compile 1836 videos featuring neutral expressions and frontal views to create the MEAD-Neutral dataset, in line with the methodology in (Zhang et al., 2023). From this dataset, we randomly choose 240 videos across 6 identities for testing. At testing stage, we employ a protocol that mirrors real-world usage, where the video and audio are from different sources, and the reference image is taken from the current frame. This is consistent with the unpaired evaluation protocol used by Wav2Lip (Prajwal et al., 2020b) and VideoRetalking (Cheng et al., 2022), ensuring a fair comparison.

**Evaluation Metrics.** The experiments are designed to assess the method's visual fidelity, identity preservation, and lip synchronization capabilities. Frechet Inception Distance (FID) (Heusel et al., 2017) is employed for visual quality assessment. We choose FID because this task lacks pixel-level ground truth; the input mouth regions are altered, making pixel-level metrics like PSNR, SSIM, and LPIPS less suitable. FID measures the similarity between the distribution of generated images and real images, providing a robust metric for visual fidelity in the absence of ground-truth talking videos. Identity preservation is measured by calculating the cosine similarity (CSIM) between the identity embeddings of the source and generated images. Lip synchronization is evaluated using lip-sync-error confidence (LSE-C) (Prajwal et al., 2020b).

**Compared Baselines.** The proposed method is benchmarked against several state-of-the-art real-time video dubbing techniques: 1) Wav2Lip (Prajwal et al., 2020b), which is renowned for generating realistic lip synchronization in videos by utilizing a robust pre-trained lip-sync discriminator; 2) VideoRetalking (Cheng et al., 2022), which delivers high-quality audio-driven lip synchronization for talking head video editing through a process of expression neutralization, lip-sync generation, and identity-aware enhancement; 3) DI-Net (Zhang et al., 2023), which is noted for creating photo-realistic and emotionally consistent talking face videos by employing a dual-encoder framework coupled with a facial action unit system; 4) TalkLip (Wang et al., 2023), which introduces an innovative contrastive learning approach to improve lip-speech synchronization and utilizes a transformer to encode audio in sync with video, taking into account the global temporal dependencies of audio.

## 4.2 QUANTITATIVE EVALUATION

Table 1 presents the results of our quantitative analysis on the HDTF and MEAD-Neutral datasets. MuseTalk outperforms the competition, achieving the highest scores in FID and CSIM, and comparable results in LSE-C. As for visual quality, Wav2Lip, VideoRetalking, and TalkLip, are all trained on resized face regions of $96 \times 96$ pixels, suffer from lower clarity, as indicated by lower FID scores. Even with the DI-Net reproduction, directly training a high-resolution Wav2Lip (Wav2Lip-192) did not improve clarity, showing worse results than Wav2Lip-96. On the other hand, DI-Net achieves the second-best FID and CSIM scores in the HDTF dataset because it leverages a deformation-based method that preserves high-frequency texture details, excelling in clarity. Nevertheless, its method of randomly sampling a list of reference images introduces redundant information to the model, which highly restricts natural lip movement and therefore sacrifices some lip-sync accuracy, as shown in the LSE-C score. Furthermore, the DI-Net model's performance on the MEAD dataset is unsatisfactory, with the generated results exhibiting noticeable color differences, likely due to its inpainting module overfit to the HDTF dataset. In contrast, MuseTalk, combined with our proposed SIS reference image sampling method and multi-scale data fusion structure, outperforms other baseline methods across FID and CSIM scores in both the HDTF and MEAD-Neutral datasets. As for audio-visual synchronization, our method underperforms TalkLip and DI-Net in LSE-C score but is only slightly lower than Wav2Lip and VideoRetalking. We attribute this to the AAM strategy for

---

[2]https://huggingface.co/stabilityai/sd-vae-ft-mse

Table 1: Performance Metrics for HTDF and MEAD-Neutral. The best results are shown in **bold** and second best results are highlighted with underlined font. IMP shows the improvement of MuseTalk over the best model.

| | HTDF | | | MEAD-Neutral | | |
|---|---|---|---|---|---|---|
| | FID↓ | CSIM↑ | LSE-C↑ | FID↓ | CSIM↑ | LSE-C↑ |
| Wav2Lip (Prajwal et al., 2020b) | 11.21 | 0.8184 | 7.46 | 24.11 | 0.8432 | **6.74** |
| VideoRetalking(Cheng et al., 2022) | 10.93 | 0.7989 | **7.7** | 26.82 | 0.8380 | 5.82 |
| TalkLip (Wang et al., 2023) | 17.61 | 0.7898 | 2.29 | 45.28 | 0.8509 | 2.07 |
| DI-Net (Zhang et al., 2023) | 7.27 | 0.8154 | 6.17 | 26.34 | 0.8274 | 4.4 |
| Ground Truth | 0.00 | 1.0 | 7.81 | 0.00 | 1.0 | 7.25 |
| Ours | **6.43** | **0.8225** | 6.53 | **13.42** | **0.8662** | 5.15 |
| IMP | 11.55% | 0.87% | -15.19% | 44.34% | 1.8% | -23.59% |

Lip-sync loss optimization, which enhances the mutual information between audio and visual correlation. However, since Wav2Lip and VideoRetalking are specifically designed to directly optimize the LSE-C metric, they achieve higher scores but at the cost of visual quality. Overall, our approach prioritizes visual quality and facial preservation, aiming for optimal performance across all metrics.

## 4.3 QUALITATIVE EVALUATION

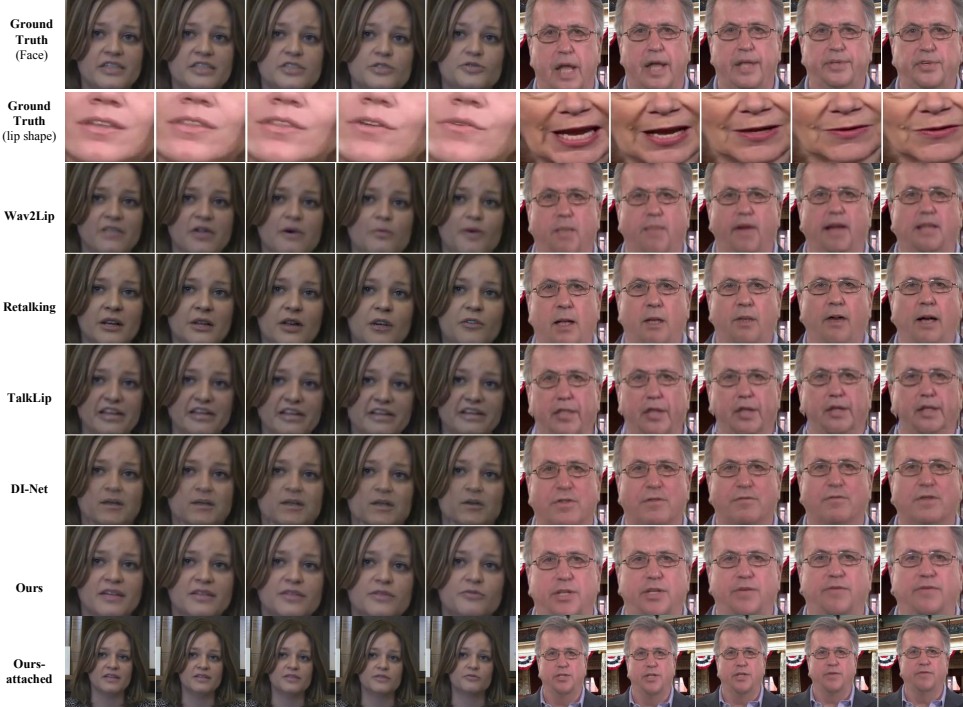

Figure 3: Qualitative comparisons on HDTF dataset with state-of-the-art methods are presented here (zoom in for finer details). The top two rows illustrate the input video frames paired with the corresponding the edited audio, where the lip shapes of the faces are used to visually represent the input audio.

To facilitate a direct visual comparison among the methods under evaluation, several illustrative examples are included in Fig. 3. Upon examination, it is evident that methods such as Wav2Lip

Table 2: Ablation study on sampling method. The best results are shown in **bold**.

| Sampling strategy | FID↓ | CSIM↑ | LSE-C↑ |
|---|---|---|---|
| Random Sampling | 23.95 | 0.7576 | 5.46 |
| Distinct-Mouth Sampling (Lip Motion Dissimilarity Set) | 7.58 | 0.7994 | **6.95** |
| Pose-Aligned Sampling (Pose-Aligned Image Set) | **4.39** | **0.8372** | 1.71 |
| Selective Information Sampling | 6.43 | 0.8225 | 6.53 |

Table 3: Ablation study on multi-scale fusion. The best results are shown in **bold**.

| Configuration | FID↓ | CSIM↑ | LSE-C↑ |
|---|---|---|---|
| *w/o* multi-scale fusion | 10.52 | 0.8072 | 2.17 |
| *w* partial multi-scale fusion | 10.69 | 0.7961 | 4.10 |
| Complete (multi-scale fusion) | **6.43** | **0.8225** | **6.53** |

- *w/o multi-scale fusion* indicates that cross attention is used only in the first layer of Unet.
- *w partial multi-scale fusion* indicates that cross attention is used in the first two layers of Unet.
- *Complete* indicates that cross attention is used in all layers of Unet.

and TalkLip often produce synthesized mouth regions that appear blurry. VideoRetalking results in jagged artifacts around the lip area and overly smooths the face region. DI-Net, in particular, induces noticeable changes in the subject's identity within the generated results, although it maintains better facial clarity. Our proposed method, however, stands out by delivering superior performance in both visual quality and identity consistency. From the last row of Fig. 3, it is clear that our method integrates seamlessly with the original image, leaving no visible traces. Additional video results and a comprehensive user study evaluation are provided in the supplementary materials.

## 4.4 ABLATION STUDIES

To evaluate the impact of each main component used in MuseTalk, we make an comprehensive ablation study experiment on HDTF dataset.

**Selective Information Sampling.** We analyze the impact of different sampling methods. As shown in Table 2, random sampling approaches, previously used in studies such as (Zhang et al., 2023; Cheng et al., 2022; Prajwal et al., 2020b), result in lower-quality images, reflected in higher FID scores. In contrast, we propose two alternative sampling methods: Distinct-Mouth Sampling and Pose-Aligned Sampling, to demonstrate the effectiveness of our Selective Information Sampling (SIS) strategy in filtering out redundant information. Distinct-Mouth Sampling, which emphasizes capturing detailed lip movements, improves the lip synchronization score (LSE-C), but compromises visual fidelity. On the other hand, Pose-Aligned Sampling focuses on matching head poses, achieving the highest image quality with the lowest FID and highest CSIM scores. However, this approach sacrifices lip sync accuracy, as the model reproduces more reference information instead of generating authentic lip movements, leading to an information leakage issue. Our Selective Information Sampling method strikes a balance by aligning head poses with the target while preserving distinct lip movements. This dual focus allows the model to capture intricate mouth details, resulting in superior performance across all metrics, balancing both image quality and lip sync accuracy.

**Multi-scale Fusion.** Leveraging the multi-scale data fusion mechanism within the UNet architecture, MuseTalk achieves effective audio-visual integration for visual dubbing. As shown in Table 3, shallow feature fusion proves insufficient, particularly in enhancing lip synchronization (LSE-C). In contrast, the full multi-scale fusion significantly improves both audio-visual coherence and image quality, underscoring its importance in achieving high-quality results.

**Adaptive Audio Modulation.** As shown in Table 4, the addition of $\mathcal{L}_{sync}$ leads to a significant average improvement across the three evaluation metrics (FID, CSIM, LSE-C) for varying audio

Table 4: Ablation study on adaptive audio modulation. The best results are shown in **bold**.

| $\mathcal{L}_{sync}$ Version | Audio Segment Length T | FID↓ | CSIM↑ | LSE-C↑ |
|---|---|---|---|---|
| | T = 1 | 9.64 | 0.8068 | 4.37 |
| | T = 3 | 9.32 | 0.8049 | 4.53 |
| w/o $\mathcal{L}_{sync}$ | T = 5 | 9.38 | 0.8089 | 4.18 |
| | T = 7 | 9.59 | 0.8042 | 3.13 |
| | T = 9 | 9.68 | 0.8088 | 2.85 |
| | T = 1 | 6.88 | 0.8239 | 5.85 |
| | T = 3 | 6.92 | **0.8239** | 6.09 |
| w $\mathcal{L}_{sync}$ | T = 5 | **6.43** | 0.8225 | **6.53** |
| | T = 7 | 6.92 | 0.8248 | 4.92 |
| | T = 9 | 7.14 | 0.8244 | 3.33 |
| Avg IMP* | - | 27.97% | 2.13% | 40.19% |
| Wav2lip(Prajwal et al., 2020a) | T = 5 | 10.24 | 0.7876 | 5.32 |

\* Avg IMP denotes the average improvement with or without lip-sync loss on three evaluation metrics (FID: X%, CSIM: Y%, LSE-C: Z%).

segment lengths. This demonstrates that $\mathcal{L}_{sync}$ helps the model extract more relevant information from both modalities, optimizing their mutual information. Moreover, the input information volume hasn't been thoroughly evaluated. We use $T$ to represent the number of audio segments per video frame, with T=1 corresponding to 40 ms at 25 FPS. The results show that without $\mathcal{L}_{sync}$, longer audio segments (T=1,3,5) have minimal impact on LSE-C. In contrast, incorporating $\mathcal{L}_{sync}$ increases LSE-C as $T$ grows, peaking at 6.53 for T=5, demonstrating $\mathcal{L}_{sync}$'s ability to modulate relevant information from larger input volumes. However, when $T > 5$, LSE-C decreases sharply (as shown by the red values in Table 4), likely due to irrelevant information dominating the input, creating noise that disrupts the effectiveness of $\mathcal{L}_{sync}$.

Besides, we also identify an issue with the training of SyncNet in previous work (Prajwal et al., 2020a). As shown in the last row of Table 4, although it achieves a higher LSE-C value, it significantly impacts clarity (FID and CSIM). We argue that the contrastive learning strategy of SyncNet may be capturing redundant information, such as the correlation between user identity in facial images and audio features. This negatively affects the model's ability to extract texture information from the reference image. To address this problem, we improve the training strategy by sampling from only one individual within a batch to create positive and negative samples. This modified SyncNet training approach can enhance both clarity and audio-visual consistency simultaneously.

## 4.5 CONCLUSION

We propose MuseTalk, a groundbreaking framework designed to address the significant challenges in generating high-quality, real-time talking faces. Inspired by Stable Diffusion, we bypass the time-consuming diffusion process and directly model the correlation between audio and visual data, utilizing a novel combination of a Variational Autoencoder (VAE) and Whisper models for feature extraction, alongside a U-Net architecture for generating targets in latent space. This innovative approach not only ensures high resolution and identity consistency but also achieves precise lip-speech synchronization. To be specific, MuseTalk integrates audio features and latent visual representations through multi-scale feature fusion within the U-Net, optimizing the output results through the proposed SIS and AAM strategies. The experiments conducted on the HDTF and MEAD-Neutral datasets show the superior performance of MuseTalk compared to existing state-of-the-art methods, underscoring its potential to set new standards in digital communication and multimedia. Besides, MuseTalk can achieve a generation speed of 30 frames per second on an NVIDIA Tesla V100, confirming its capability for real-time applications.

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

# A APPENDIX

This Appendix provides additional information that could not be included in the main manuscript due to space constraints. First, we present further experimental results, including a well-designed user study and additional visualizations. Second, we offer a discussion on the limitations of MuseTalk and outline potential future research directions.

## A.1 USER STUDY

To assess the quality of lip synchronization, human judgment is relied upon. A user study was conducted to further evaluate the performance of our proposed method. For this study, talking head videos were created by different methods using 55 unsynced audio-video pairs from the HDTF datasets. Ten participants were asked to rate each video based on visual quality and lip-sync accuracy. They were provided with a five-point scale (with 1 being the lowest and 5 being the highest) for their evaluations. A total of 550 ratings were collected. As indicated in Table 5, the majority of participants awarded higher scores to our method in terms of visual quality, lip-sync quality, and identity consistency. More visualizations are shown in Fig. 4.

Table 5: User Study. The best results are shown in **bold** and second best results are highlighted with underlined font. IMP shows the improvement of MuseTalk over the best model.

| Method | Visual Quality↑ | Identity Consistency↑ | Lip-Sync Quality↑ |
|---|---|---|---|
| Wav2lip (Prajwal et al., 2020b) | 1.87 | 2.86 | 2.69 |
| VideoRetalking (Cheng et al., 2022) | 2.16 | 2.90 | 3.20 |
| DINet (Zhang et al., 2023) | 3.45 | 2.85 | 3.15 |
| Ours | **3.62** | **3.55** | **3.41** |
| IMP | 4.92% | 22.41% | 6.56% |

## A.2 LIMITATION AND FUTURE WORK

While MuseTalk demonstrates a notable improvement in face region resolution (256x256) compared to other state-of-the-art methods, it has yet to reach its full resolution potential. Additionally, certain facial details—such as mustaches, lip shape, and color—are not always well-preserved, which can affect identity consistency. Lastly, occasional jitter is observed due to the single-frame generation process, compromising smoothness.

To address these limitations, future work will focus on incorporating higher-quality training data and integrating a temporal module to reduce jitter and ensure smoother transitions. These enhancements aim to improve both resolution and overall visual consistency. Moreover, incorporating super-resolution models like GFPGAN as a post-processing step could further elevate output quality in real-world applications.

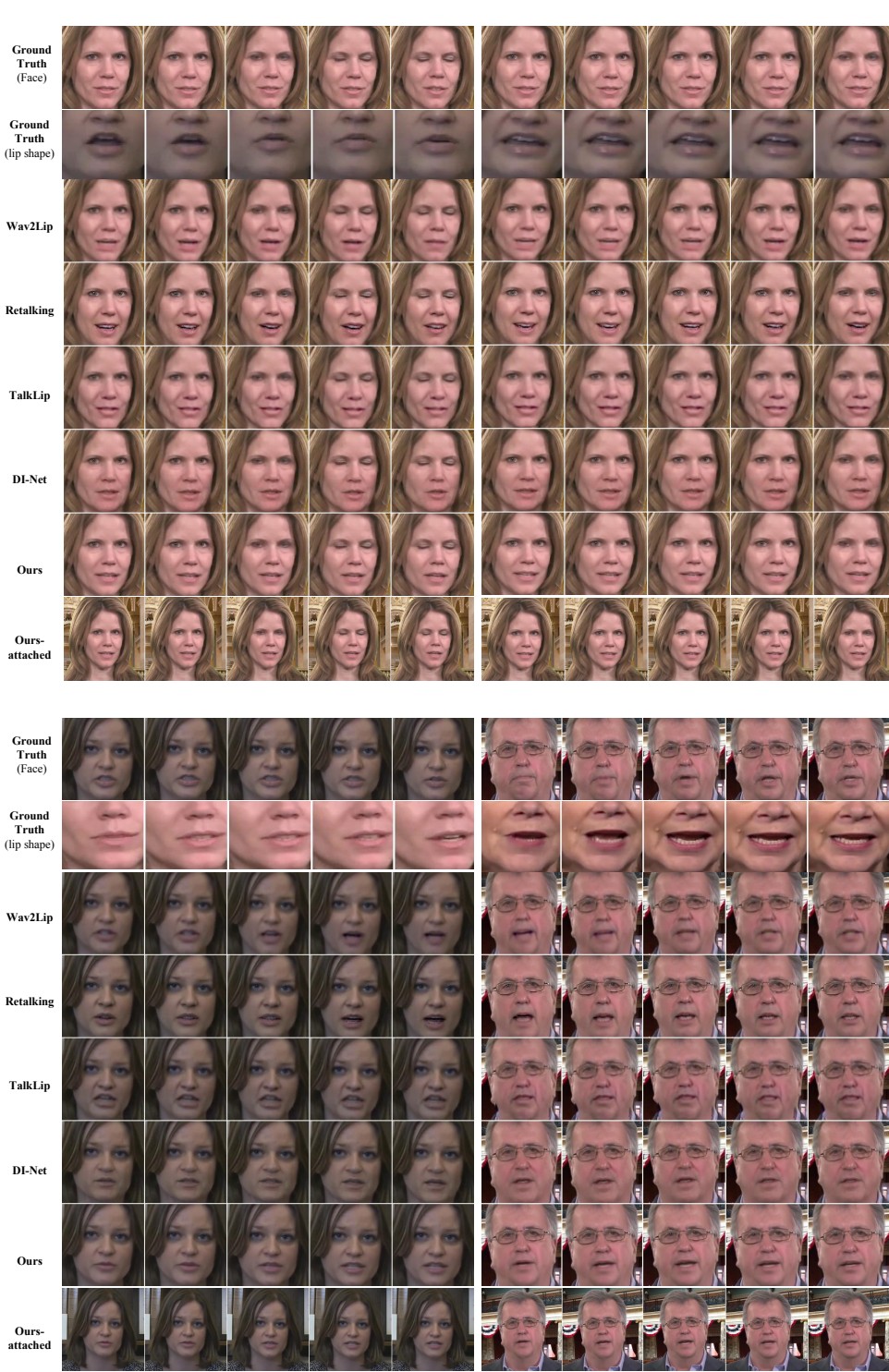

Figure 4: Qualitative comparisons on HDTF dataset with state-of-the-art methods are presented here (zoom in for finer details). The top two rows illustrate the input video frames paired with the corresponding the edited audio, where the lip shapes of the faces are used to visually represent the input audio.

