# OpenReview forum: "MuseTalk: Real-Time High Quality Lip Synchronization with Latent Space Inpainting"
_ICLR.cc/2025/Conference — ICLR 2025 Conference Withdrawn Submission_

### Official Review · Reviewer_eTic · 2024-10-30

**Soundness:** 1
**Presentation:** 2
**Contribution:** 2
**Rating:** 3
**Confidence:** 5

**Summary:**

MuseTalk is developed to deliver precise lip synchronization for real-time applications. By mapping lip-sync targets into the latent space of a VAE and employing a U-Net architecture to blend audio and visual features, MuseTalk achieves high-quality talking face video generation. It enables real-time output at a 256x256 resolution with a frame rate of 30 FPS.

**Strengths:**

- The main ideas are clearly presented throughout the paper.
- The evaluations are thoroughly conducted with the proposed modules.
- Code is provided.

**Weaknesses:**

- L197: I do not see the two adjustments as novel. 1) “bypass the complex and time-consuming diffusion process” means you opt for a GAN-based model. People try diffusion-based methods for their better generative capacity, not for their speed. In another hand, GAN-based Lip-Sync methods have been widely explored [1,2,3,4], which are not discussed in this paper. 2) Using an occluded lower half face input is also a common practice in Lip-Sync, I do not see it is an adjustment in any way.


- L294: “However, achieving precise synchronization is challenging due to different sampling rates between the two modalities. While our multi-scale fusion architecture addresses this issue to some extent, it does not entirely resolve it.“  Cannot understand why the sampling rate between the two modalities affects audio-visual synchronization? "multi-scale fusion" was also explored in previous works like wav2lip, [1,2,3,4].

- Fig. 2 (b) I couldn’t understand what was meant by “information volume” and “information modulator” until realizing that they are simply terms referring to the temporal length and SyncLoss. It seems these are just “fancier” expressions for familiar concepts.

- I would try to refine the figures. Sometimes find too small the text and face.

- One contribution claimed by the authors is to improve the lip-sync accuracy. However, from the evaluation in Table 1, I see only a compromise between lip-sync and generative quality.

- From the supp files, the mouth region appears noticeably more blurred than other areas of the face.

[1] RADIO: Reference-Agnostic Dubbing Video Synthesis

[2] Dubbing for Everyone: Data-Efficient Visual Dubbing using Neural Rendering Priors

[3] Audio-driven Talking Face Generation with Stabilized Synchronization Loss

[4] StyleSync: High-Fidelity Generalized and Personalized Lip Sync in Style-based Generator

**Questions:**

- Regarding the classification of related methods, I don’t view methods like Wav2Lip as “few-shot” since they can directly handle unseen identities without finetuning. Could you clarify why it is referred to as few-shot face visual dubbing?

- L331: is there any identity overlay between the 20 testing videos and the training videos?

---

### Official Review · Reviewer_SCt4 · 2024-11-01

**Soundness:** 2
**Presentation:** 2
**Contribution:** 2
**Rating:** 5
**Confidence:** 5

**Summary:**

This paper proposes MuseTalk, which generates lip-sync targets in a latent space, enabling high-fidelity talking face video generation with efficient inference. From the innovation perspective, this work resembles more of a technical report, as the designs introduced are largely existing engineering implementations. In terms of performance, there is no significant improvement on image quality and audio-lip synchronization compared to other methods.

**Strengths:**

1. This paper removes the denoising process of the conditional latent diffusion model and introduces the spatial efficiency of VAE latent space along with the UNet architecture. This idea is reasonable and solid.
2. The paper's exploration of how to select reference images to ensure generation quality is meaningful and provides valuable insights that can be referenced by other works.
3. The open-source code provided in this paper is complete and easy to reproduce.
4. The writing of this paper is clear, and the description of the methods is easy to follow.

**Weaknesses:**

* Method Description
    1. Please explain how the channel dimension is transformed from 2c to c after concatenating the reference image and the source image along the channel dimension.
    2. What does the $t$ in $I^t_{ref}$ represent? If it does not refer to the diffusion model, should $t$ be removed to avoid confusion?
    3. What should be done if SIS selects multiple reference images? According to Figure 1, there should be only one reference image for each source image.
    4. The inspiration for AAM lacks logical coherence and supporting references. What are the differences between AAM and directly using a 0.2-second segment, and how do they compare in terms of performance?
* Performance Evaluation
    1. This paper removes the denoising manner but does not analyze the impact of this removal, particularly on image quality and lip sync. There are already related papers, such as DiffTalk [1].
    2. The baselines compared in the paper are relatively weak. It would be beneficial to include a comparison with stronger baselines like IP_LAP [2], which is known for its stable performance.
    3. The paper conducts experiments solely on the HDTF dataset, which features relatively simple scenes with limited head and mouth movements. It would be advisable to consider testing on more complex datasets, such as VoxCeleb and VFHQ.
    4. The title of the paper includes "real-time," but the comparative evaluation only addresses quality and does not provide any information about inference speed.

[1] S. Shen et al., "DiffTalk: Crafting Diffusion Models for Generalized Audio-Driven Portraits Animation," IEEE/CVF Conference on Computer Vision and Pattern Recognition (CVPR), 2023, pp. 1982-1991.

[2] W. Zhong et al., "Identity-Preserving Talking Face Generation with Landmark and Appearance Priors," IEEE/CVF Conference on Computer Vision and Pattern Recognition (CVPR), 2023, pp. 9729-9738.

**Questions:**

Please consider addressing or supplementing the relevant experiments based on the points mentioned in the weaknesses.

---

### Official Review · Reviewer_SMyG · 2024-11-03

**Soundness:** 3
**Presentation:** 3
**Contribution:** 2
**Rating:** 3
**Confidence:** 4

**Summary:**

This paper uses the idea of latent space inpainting to obtain real-time lip synchronization. They advocate a frozen encoder to obtain a latent embedding for a reference face and the occluded face that needs to be inpainted while providing an audio embedding using Whisper model to a UNet that is trained using a set of losses. The main contribution of the work is the engineering involved in designing the sampling of the reference frame that ensures consistent pose while having differing lip-shape and the effort to obtain the correct length while synchronization. Conceptually, the paper builds up on existing architectures and works.

**Strengths:**

1. The efforts involved in analysing the role of sampling of the reference frame and adaptation of synchronization parameters is useful. The method is useful in obtaining real-time lip synchronization for talking faces.
2. The ablation analysis for each of the three contributions - sampling, multi-scale fusion and synchronization strategies is useful

**Weaknesses:**

1. The work does not provide baseline comparisons with many of the existing efforts that have been published recently such as V-Express, Audio-driven Talking Face Generation with Stabilized Synchronization Loss - ECCV 2024, StyleSync - CVPR 2023, Resyncer - ECCV 2024. Besides there are efficient high speed talking face works that use Gaussian Splatting such as FlashAvatars that provide higher quality with higher inference speeds. In view of this, it is not clear that the proposed method provides improved results or insights over the existing works
2. The proposed work does not provide significant insights into high-quality talking face synthesis. The efforts, while showing good results are similar to many existing works in terms of architecture and loss functions. The main improvements appear to be in terms of adaptation of sampling and synchronization strategies
3. The proposed method would not adapt to individual idiosyncracies - such as enunciation of vowels or style of speaking. As such, this would limit the method from being adopted for high-quality applications such as entertainment
4. The method uses a per-frame prediction strategy. This can be limiting as identified by the authors. Particularly when it comes to talking faces with people moving pose while talking, it can be challenging. The results do not demonstrate the extent to which this is limiting by evaluating the method on challenging scenarios.

**Questions:**

1. Could the authors provide comparisons with existing state of the art techniques that validates the quality of the results against current state of the art. At present the comparisons provided are rather limited. Else, proper justification is required why the current state of the art techniques are not applicable and the chosen set of baselines suffice. Further there are many commercial offerings that demonstrate high-quality lip-sync for instance by HeyGen. How does the proposed method compare against existing commercial offerings in this space?
2. How would the authors ensure that the style and manner of person specific speaking is ensured?
3. Can the authors consider more challenging, real-world scenarios and confirm that their method indeed generalizes to more challenging scenarios

**Details Of Ethics Concerns:**

The proposed techniques would as they stand violate the EU AI Act and the authors would need to confirm that they will watermark any results to ensure that the generated samples confirm to the EU AI Act.

---

### Official Review · Reviewer_qLZa · 2024-11-06

**Soundness:** 2
**Presentation:** 2
**Contribution:** 2
**Rating:** 5
**Confidence:** 4

**Summary:**

This work addresses the few-shot face visual dubbing problem, i.e., regenerate mouth region according to the given audio. The goal is to achieve high-resolution, identity-consistency, and high lip-sync quality. The pipeline follows a SD-like framework (VAE & diffusion UNet) with conditioning on audio input. To bridge the gap between training and inference on head pose, the authors propose Selective Information Sampling to select the reference images with head poses aligned to the target. Experiments are conducted on HDTF and MEAD.

**Strengths:**

The authors give detailed description of the framework (e.g., the loss function), though there are many confusions, e.g., what the meaning of 'bypass the complex and time-consuming diffusion process'?

It seems the proposed model achieves slightly better results than previous work. But it suggests to provide more examples to avoid cherry pick.

Selective Information Sampling is proposed to sample the frames with aligned pose but different mouth shapes, it is reasonable to me.

**Weaknesses:**

The SD-like framework is not novel to me given lots of previous works [1][2]. Beyond this, the authors propose two strategies to further improve the performance: Selective Information Sampling & Adaptive Audio Modulation. SIS is reasonable to me but the second leverages lip-sync loss which is first proposed in 2020, making the contribution of this work lighter.

It is not clear that how do you determine the k in SIS. Both Pose-Aligned Image Set and Lip motion Dissimilarity Image Set use the same hyperparameters, what if there is no overlap between them?

The results show that the proposed method achieve better performance on FID but worse on lip sync. Also, in table 2, SIS improves the FID but at the cost of lip sync quality compared to Lip Motion Dissimilarity Set.

[1] EMO: Emote Portrait Alive - Generating Expressive Portrait Videos with Audio2Video Diffusion Model under Weak Conditions

[2] Hallo: Hierarchical Audio-Driven Visual Synthesis for Portrait Image Animation

**Questions:**

How do you determine the k in SIS?

Can SIS be applied to other schemes?

---

### Note · Authors · 2024-11-13

I have read and agree with the venue's withdrawal policy on behalf of myself and my co-authors.